# Experimental and Numerical Simulation Study of Devolatilization in a Self-Wiping Corotating Parallel Twin-Screw Extruder

**DOI:** 10.3390/polym12112728

**Published:** 2020-11-17

**Authors:** Masatoshi Ohara, Yuya Sasai, Sho Umemoto, Yuya Obata, Takemasa Sugiyama, Shin-ichiro Tanifuji, Shin-ichi Kihara, Kentaro Taki

**Affiliations:** 1Department of Natural Science, Graduate School of Kanazawa University, Kakuma machi, Kanazawa, Ishikawa 920-1192, Japan; ohara.masatoshi@shibaura-m.com (M.O.); ume.purple@gmail.com (S.U.); obataidy@stu.kanazawa-u.ac.jp (Y.O.); 2Shibaura Machine, Oooka, Numazu, Shizuoka 410-8510, Japan; sasai.yuya@shibaura-m.com (Y.S.); sugiyama.takemasa@shibaura-m.com (T.S.); 3HASL, Shakuji machi, Nerimaku, Tokyo 177-0041, Japan; tanifuji@hasl.co.jp; 4Graduate School of Advanced Science and Engineering, Hiroshima University, Kagamiyama, Higashi Hiroshima, Hiroshima 739-8527, Japan; snkihara@hiroshima-u.ac.jp; 5School of Mechanical Engineering, Kanazawa University, Kakuma machi, Kanazawa, Ishikawa 920-1192, Japan

**Keywords:** self-wiping, devolatilization, toluene, polypropylene, twin-screw extruder, 2.5D Hele–Shaw flow, simulation

## Abstract

Devolatilization is an important process for separating and removing unnecessary residual volatile substances or solvents during the production of polymers using twin-screw extruders. Latinen proposed a surface renewal model to determine the concentration of volatile components in the extrudate of a single-screw extruder. When a twin-screw extruder is used to calculate the concentration, it is necessary to use the exposed surface area of the resin in the starved region of Latinen’s model, which, however, is difficult to estimate. In our previous work, we numerically determined resin profiles of the screws using the 2.5D Hele–Shaw flow model and the finite element method, which helps in estimating the surface area of devolatilization. In this study, we numerically analyzed the volatile concentration of the extrudate in a self-wiping corotating twin-screw extruder using Latinen’s surface renewal model along with our resin profile calculation method. The experimental results of the concentrations of the volatile component (toluene) in the extrudate of polypropylene agreed well with its numerical calculation with a relative error of 6.5% (except for the data of the lowest rotational speed). Our results also showed that decreasing the flow rate and increasing the pump capacity were effective for removing the volatile component. The screw pitch of a full-flight screw was not affected by the devolatilization efficiency with a fixed flow rate and screw speed.

## 1. Introduction

A twin-screw extruder is a versatile polymer-processing machine for pelletizing, blending, alloying, compounding, devolatilizing, performing chemical reactions, and forming dies of molten polymers [1]. During the manufacture of polymers, it is important to separate and remove unnecessary residual volatile organic compounds (VOCs) such as solvents, monomers, and other by-products [1], as doing so improves the product quality and minimizes the chances of coming in contact with a volatile component. This process of removing volatile components is called devolatilization. Twin-screw extruders can be used to continuously carry out devolatilization with high productivity and a small footprint. Devolatilization has been investigated using various screw configurations and operating parameters to increase the production rate and improve the quality of polymers. Numerical simulation of devolatilization using twin-screw extruders is a promising method to reduce the number of actual experiments required for product development.

The following devolatilization phenomena—flash, surface renewal, and foaming—can be performed using twin-screw extruders.
Flash devolatilization: A high-temperature and high-pressure polymer solution is first fed into an extruder. The high pressure is then released such that the extruder is at atmospheric pressure. As a result, the volatile component can be separated from the solution and removed through a process similar to flash distillation. Thereafter, the volatile components are discharged from a vent upstream of the extruder. The devolatilization efficiency of flash devolatilization can be determined using the vapor–liquid diagram of the volatile component at the initial and atmospheric-pressure levels. Although flash devolatilization is an essential part of the devolatilization sequence, only a few studies have investigated this process [2].Surface renewal devolatilization: This method depends on the surface area and residence time of the resin and the mass transfer rate of the volatile component. At a vent downstream of the feed port, the volatile substance diffuses out from the surface of the polymer solution owing to the difference between the partial pressure of the volatile component in the polymer solution and the atmospheric pressure. Latinen presented a pioneering work on modeling the surface renewal devolatilization phenomena in a single-screw extruder [3]. Surface renewal involves distributive mixing between the barrel and the screw, which enhances the mass transfer of the volatile component. His model uses the surface area, residence time, and mass transfer coefficient to calculate the devolatilization efficiency. The model has been extended to counterrotating and corotating twin-screw extruders. It has also been used for experimental evaluation of various types of screws and chemicals [4,5,6,7,8,9,10,11,12].Foam devolatilization: When the residual volatile components are present in the range of a few parts per million, removal becomes very difficult solely by increasing the degree of vacuum through surface renewal devolatilization. To overcome this shortcoming, water is injected into the resin to induce steam-bubble foaming. The volatile component of the resin diffuses out into the steam bubbles. When the bubbles collapse on the resin surface, the volatile component of the bubbles dissipates in air. Because the total surface area of the bubbles is much larger than that of the surface renewal devolatilization area, the overall mass transfer increases significantly. This water-assisted foam devolatilization method is efficient in removing low-concentration volatile components [2].

Hence, it is clear that for modeling the devolatilization phenomenon, it is essential to design a function of the surface area, residence time, and mass transfer rate of the volatile component. The mass transfer rate depends on the physicochemical properties of the resin and volatile components. The surface area and residence time change depending on the degree of fill, which is a function of the flow rate, screw speed, and screw geometry.

Wang and Hashimoto derived the equations of surface area and residence time for a corotating twin-screw extruder [8]. They elaborated their expressions and method for treating the surface area. However, the complexity of their expressions makes it difficult to calculate devolatilization in practical applications. Therefore, it is necessary to develop a simple and versatile calculation method for both the surface area and residence time for application in industry. To date, however, no major study has been performed to realize the accuracy and usefulness of the devolatilization model, although, very recently, Hirschfeld and Wünsch studied the mass transfer of devolatilization in a bubble-free polymer solution by focusing on the surface renewal and mixing effect. They used computational fluid dynamics (CFD) to quantify the surface area for devolatilization using OpenFOAM^®^ [13]. Their study paves the way to calculate the surface area using the volume-of-fluid (VOF) method without the complicated expressions of screw geometry. However, the calculation cost is very high, which limits the process scale-up to whole-screw calculation, thereby limiting its application in the industrial twin-screw extrusion.

Recently, our group developed a calculation method for the resin profile on starved or unfilled screws of a self-wiping corotating twin-screw extruder [14] using the 2.5D Hele–Shaw flow model and the finite element method. Our calculation method uses the screw geometry, resin viscosity, feed rate, screw speed, and temperature settings. The screw geometry is captured as a height profile of each element of the finite element method. This gives us the resin profile of the starved screw surface on not only the machine direction but also the circumference direction of the entire screw element. Because the flow velocity in the normal direction to the flow channel is neglected, the number of variables is reduced by four times. The calculation time is short at 10 min. Hence, the resin profile is promising for calculating the area of surface renewal devolatilization.

In this paper, we established the numerical calculation method for the volatile concentration of the extrudate in a self-wiping corotating twin screw extruder using the 2.5D Hele–Shaw flow model and the finite element method. Toluene was used as an example volatile component in the extrudate of polypropylene and its concentration was experimentally measured in a vacuum-vent-equipped self-wiping corotating twin-screw extruder. The experimental results agreed well with the simulation results except for the case of the lowest rotation speed (60 rpm). We also reviewed several theoretical and experimental studies to elucidate the aim of our study.

The paper is organized as follows. In Section 2, we reviewed Latinen’s surface renewal model and derived a simple formula for determining the volatile concentration, neglecting the diffusion in the screw axial direction. The surface boundary length in the formula was evaluated using the calculation scheme that we proposed in an earlier work [14]. We showed an algorithm for calculating the volatile concentration. In Section 3, we described the experiments conducted to investigate the devolatilization. We used homopolypropylene as the resin and toluene as the volatile component. The vaporized toluene was collected from the vent located downstream and was cooled to liquefy. The concentration of toluene in the extrudate was estimated from the mass of the condensed toluene. We examined the volatile concentration at the screw rotation speeds of 60, 90, 120, 150, and 180 rpm with a flow rate of 3 kg/h. In Section 4, we compared the numerical and experimental results. Our simulation results agreed well with the experimental results with a relative error of 6.5%, except for the lowest screw rotation speed. We also discussed the reason for the discrepancy observed at the screw rotation speeds of 60 rpm. Section 5 summarizes the study.

## 2. Theory

Our theoretical model of surface renewal devolatilization is based on Latinen’s model [3] (Figure 1). The exposed surface boundary length, where devolatilization chiefly occurs, is marked as $\bar{abc}$ on the cylinder of a single-screw extruder.

In the starved screw of a corotating twin-screw extruder, the surface of the molten resin formed on the barrel surface promotes surface renewal devolatilization. If the mass transfer rate of the volatile component at the surface is in a steady state, the amount of the volatile component evaporated as the gas is in balance with the mass transferred from the resin (Equation (1)). The third term in Equation (1) is obtained when the mass transfer in the resin occurs in a semi-infinite medium [15].
(1)$$k_{m}\left(C-C^{*}\right)=D_{m}\left(\frac{\partial C}{\partial y}\right)=D_{m}\frac{\left(C-C^{*}\right)}{{\left(\pi D_{m}\theta \right)}^{1/2}}$$
where *k*_m_ is the mass transfer coefficient, *C* is the concentration of the volatile component in a resin, *C** is the equilibrium concentration of the volatile component at a given partial pressure of the volatile component and resin temperature, *D*_m_ is the diffusion coefficient of the volatile component in the resin, and *θ* is the exposure time. The exposure time corresponds to the time when the molten resin passes through the exposed surface boundary length, which can be evaluated using Equation (2):(2)$$\theta =\frac{\bar{abc}}{\pi D_{b}N},$$
where *D*_b_ is the barrel diameter, *N* is the screw rotational speed, and $\bar{abc}$ is the exposed surface boundary length.

The time-averaged mass transfer coefficient was obtained by integrating Equation (1) with respect to exposure time *θ* and dividing it by *θ*:(3)$${\bar{k}}_{m}=\frac{1}{\theta }\int_{0}^{\theta }k_{m}d\theta =2{\left(\frac{D_{m}}{\pi \theta }\right)}^{1/2}.$$

Assuming a steady state, the material balance in the infinitesimal distance *dz* of the axial direction of screw *z* is described by the advection diffusion equation shown below:(4)$$A\frac{\partial C}{\partial t}+Au_{z}\frac{\partial C}{\partial z}=AE\frac{\partial^{2}C}{\partial z^{2}}-{\bar{k}}_{m}S\left(C-C^{*}\right),$$
where *u_z_* is the flow velocity of the screw axis *(**z*), *E* is the diffusion coefficient of the volatile component in the *z*-direction, *S* is the exposed surface boundary length, and *A* is the cross-sectional area of the resin. In Equation (4), we have neglected the flow and diffusion of the volatile component along the circumference direction of the screws.

The process is in a steady state and the convection term is dominant rather than the diffusion term of Equation (4).
(5)$$u_{z}\frac{dC}{dz}=-\frac{{\bar{k}}_{m}S}{A}\left(C-C^{*}\right).$$

The mean flow velocity *u_z_* is obtained from the mass flow rate and density of the resin:(6)$$u_{z}=\frac{Q_{w}}{\rho_{m}A},$$
where *Q*_w_ is the mass flow rate and *ρ*_m_ is the density of the resin. Dividing Equation (5) with *u_z_* and substituting it with Equation (6) give:(7)$$\frac{dC}{dz}=-\frac{{\bar{k}}_{m}S\rho_{m}}{Q_{w}}\left(C-C^{*}\right).$$

Devolatilization efficiency χ is obtained when Equation (7) is integrated between *z*_s_ and *z*_e_ by assuming that *S* is constant if the integration range is sufficiently small.
(8)$$\chi =\frac{C(z_{s})-C^{*}}{C(z_{e})-C^{*}}=\exp\left(\frac{{\bar{k}}_{m}S\rho_{m}(z_{e}-z_{s})}{Q_{w}}\right).$$

Assuming the exposed surface boundary length $\bar{\mathrm{abc}}$ is equal to *S* and combining Equations (2), (3), and (8), we obtain Equation (9).
(9)$$\chi =\frac{C(z_{s})-C^{*}}{C(z_{e})-C^{*}}=\exp\left(\frac{2{\left(D_{m}D_{b}NS\right)}^{1/2}\rho_{m}\left(z_{e}-z_{s}\right)}{Q_{w}}\right).$$

In Equation (9), the surface boundary length, *S*, is obtained from the resin surface profile on screws, which was calculated by the method proposed in our previous paper [14]. The resin profile obtained using our calculation method is shown in Figure 2.

In Figure 2d, the sum of the length of the thin layer of resin on the barrel, i.e., a-b, c-d, and e-f, corresponds to the exposed surface boundary *S*. Figure 2e shows the local degree of the fill along the barrel circumference. The exposed boundary lengths corresponding to a-b, c-d, and e-f, shown in Figure 2d, are clearly identified as the length of the zero-degree of fill. Zero-degree of fill implies that no resin was used in the simulation method to determine the resin profile. In practice, there is a clearance between the screws and the barrel, wherein the thin resin resides. According to the surface renewal model proposed by Latinen, devolatilization occurs in this thin resin layer.

If the local degree of fill is close to unity and the summation of a-b, c-d, and e-f is zero, the exposed surface boundary length *S* vanishes and the devolatilization efficiency χ becomes unity. Devolatilization does not occur in the fully filled region. Instead, it occurs in the melt-sealed zone between the screw head and the kneading disk zone, indicated as the “devolatilization zone” in Figure 2a–c. Our simulation does not consider the position of the vent port. Devolatilization occurs uniformly in the starved zone.

The volatile concentration at *z*_e_ is derived from the devolatilization efficiency:(10)$$C(z_{e})=\frac{C(z_{s})-C^{*}}{\chi }+C^{*}.$$

The resin profile in the corotating screw extruder is determined based on the input of the screw geometry, feed rate, screw rotation speed, barrel temperature, and resin properties such as viscosity function, density, thermal conductivity, and heat capacity of the polymer [14]. The height of the screw element is incorporated into the simulator. No other fitting parameter is required to determine the resin profile. Once the resin profile is determined, the volatile concentration is calculated from upstream to downstream in the starved zone during postprocessing.

The diffusion coefficient of toluene, *D*_m_, changes with the concentration of toluene in the resin and temperature. The following empirical equation is used to estimate the diffusion coefficient.
(11)$$D_{m}=a_{T}D_{r}\exp\left(\beta \frac{\varphi }{\varphi_{m}}\right)$$
(12)$$a_{T}=\exp\left[\alpha \left(T-T_{r}\right)\right],$$
where $a_{T}$ is a function representing temperature dependence, *D_r_* is a diffusion coefficient at a reference temperature, *φ* is a volatile component concentration, *φ*_m_ is the initial value of the volatile component, *T* is the temperature, *T_r_* is the reference temperature, *α* is the temperature dependence, and *β* is a model parameter that represents the concentration dependence. Diffusion coefficient *D*_r_ was determined using the values reported by Pereira et al. [16] (Table 1). Table 2 shows the parameters determined based on the experimental results. Notably, the temperatures shown in Table 1 are significantly lower than the barrel temperature of our extruder, e.g., 195 °C. The diffusion coefficient was estimated by extrapolating Equations (11) and (12).

Figure 3 shows the flow chart used for calculating the volatile concentration. The surface boundary length *S* is evaluated by the 2.5D FEM analysis. The diffusion coefficient *D*_m_ at a grid point *z_i_* can be evaluated from the temperature at *z_i_* and the volatile concentration at *z_i−_*_1_ (a previous grid point). From these values, we can obtain the devolatilization efficiency at *z_i_*. Iterating this procedure, we finally obtain the volatile concentration at *z_e_*.

## 3. Experiment

Homo-polypropylene (PP, F–704NP) was purchased from Prime Polymer (Tokyo, Japan). The melt flow rate was 0.7 g/min, measured in accordance with the ISO 1133:97 standard. The oscillation shear viscosity (complex viscosity) was measured using a modular compact rheometer (MCR 302–WESP, Anton Paar, Graz, Austria) with ϕ = 25 mm and a 1-mm-thick parallel plate fixture. The oscillation ranged from 0.01 to 100 rad/s. The strain was 1.0%. P-PTD200 and H-PTD200 temperature control units were used. The temperatures used were 180, 190, and 200 °C. Based on the Cox–Merz rule, the complex viscosity was used to obtain the parameters for the Cross model described by Equations (13) and (14), as listed in Table 3. The complex viscosity data and associated curves of the Cross model are shown in Figure 4.
(13)$$\eta \left(\dot{\gamma },T\right)=\frac{\eta_{0}}{1+{\left(\frac{\eta_{0}\dot{\gamma }}{\tau^{*}}\right)}^{\left(1-c\right)}}$$
(14)$$\eta_{0}=A\exp\left(\frac{T_{b}}{T+273.15}\right).$$

Toluene (special-grade reagent, purity 99.5%, FUJIFILM Wako Pure Chemical Corporation, Osaka, Japan) was used as a volatile component. A mixture of ethanol (reagent grade, FUJIFILM Wako Pure Chemical Corporation, Osaka, Japan) and dry ice (frozen CO_2_) was used as a cryogen to cool and condense the toluene vapor exhausted from the vent. The temperature of the cryogen was below −50 °C.

A process flow diagram of the devolatilization process is shown in Figure 5. A self-wiping corotating twin-screw extruder (L/D = 64.6, D = 26 mm, SHIBAURA MACHINE, Shizuoka, Japan) was used. Toluene was supplied at 3 wt.% (30,000 ppm) per kg PP from the injection port of the extruder using a liquid feed pump (PU-2085 Plus, JASCO, Tokyo, Japan). Vaporized toluene was collected from the vent located downstream using a vacuum pump (NVP-2100V, EYELA, Tokyo, Japan) and a vacuum trap (TR200, AS ONE, Osaka, Japan). The vacuum pressure was adjusted to an absolute pressure of 400 hPa using a needle valve unit (PSR 01, AS ONE, Osaka, Japan). A screw configuration and barrel temperature settings were given in Figure 6. The screw configuration with a starved zone length of 1024 mm was used. Table A1 shows the details of the screw elements.

The devolatilization efficiency was obtained from the mass balance of toluene.
(15)F=V+D+R,
where *F* is the flow rate of toluene injection, *V* is the flow rate of the vent, *D* is the flow rate of the extrudate, and *R* is the flow rate dissipated back to the extruder. *R* can be considered zero because the vapor concentration of toluene at the feed port was determined to be less than 1 ppm using a gas detector tube (GV 100 122L Toluene, GAS TEC, Kanagawa, Japan). Using the flow rate of toluene in the extrudate given by Equation (15), the concentration of toluene in the extrudate can be determined as follows:(16)$$C_{D}=\frac{F-V}{F-V+F_{p}},$$
where *F*_p_ is the mass feed rate of the polymer.

The mean toluene flow rate from the vent port was calculated by measuring the mass of the condensed toluene every 15 min. To evaluate the effect of the screw rotation speed on the toluene concentration at the exit of the extruder, screw rotation speeds of 60, 90, 120, 150, and 180 rpm were used.

## 4. Results and Discussion

The numerical simulation of devolatilization was performed in the starved zone by changing the screw rotational speed. Figure 7 shows the toluene concentration along the machine direction of the twin-screw extruder. The toluene concentration gradually decreased with increasing distance. The screw rotation speed of 60 rpm showed the slowest decrease, whereas that of 180 rpm showed the fastest decrease. The concentration at 60 rpm is larger than that at 180 rpm. The highest devolatilization efficiency is obtained by the fast screw rotation speed. The fast screw rotation speed gives a low degree of fill [14,17], which implies that less resin fills the screw. As the exposed surface boundary length is the length where the local degree of fill is zero, the low degree of fill increases the exposed surface boundary length, which in turn increases the devolatilization efficiency.

The simulation results of the concentrations of toluene at the extruder were compared with the experimental results. Figure 8 shows the simulation and experimental results for the toluene concentration. The experimental conditions were chosen based on the specifications of the injection pump. The maximum screw rotation speed was chosen at 180 rpm because the volatile concentration would not change at higher screw rotation speeds. The screw rotation speed was changed by 30 rpm to confirm the volatilization efficiency. The simulation and experimental results for each screw rotation speed other than 60 rpm agree well within the relative error of 6.5%. The exposed surface boundary length is predicted well using the developed devolatilization calculation method. This method can also be used to calculate the devolatilization efficiency. The deviation of the simulation results obtained at 60 rpm is greater than that determined at other speeds. The volatile concentration used for the simulation is larger than that used to obtain the experimental result. To understand the cause-and-effect relationship, we determined the resin profile using the laser-light section method, previously developed by our group [17]. Figure 9 shows that the resin profiles at 120 and 90 rpm agree well with the simulation and experimental results. Although the left side of the resin profiles at 180 and 150 rpm collapses and the simulation results do not agree completely with the experimental results, the right side of the exposed surface length agrees well with the simulation results. The worst deviation is shown by the result obtained at 60 rpm. This result is obtained because the exposed surface length of the simulation is two-third of that obtained experimentally. However, in our previous study [14], the deviation was treated as an artifact of the experiment because the resin tends to remain in the valley of the two screws when the barrel is open at slow screw speeds. Therefore, when the twin-screw extrusion is equipped with the vent port, the resin profile is expected to resemble that obtained by the laser-cross section method. Hence, the short-exposed surface length of the simulation overestimated the volatile concentration.

The resin can maintain its resin profile at a faster screw rotation speed owing to the inertia of screw rotation, while it cannot do so at slow screw rotation speeds because of gravity. More work is needed to match the simulation result with the experimental result of the resin profile at 60 rpm.

Hereafter, the developed method is applied to the flow rate, screw pitch, and equilibrium concentration, *C** (Figure 10). An increase in the flow rate increases the degree of fill, decreases the exposed surface length, and eventually decreases the volatilization efficiency. As expected and shown in Figure 10a, the volatile concentration of the extrudate increases with an increase in the flow rate.

In Figure 10b, the effect of the screw pitch is insensitive to the volatile concentration. Figure 11 shows the degree of fill distribution of 20-, 40-, and 80-mm pitches of full-flight screws at the same flow rates and screw speeds. The degree of fill of 80-mm pitch is lower than those of 20- and 40-mm pitches. It is expected that the exposed surface boundary length of 80 rpm is the largest and the devolatilization efficiency is the highest. However, the simulation results indicate that the degree of fill changed by the screw pitch does not change the volatile concentration. Because the flow rate was maintained at 3 kg/h and the screw with different pitches was homothetic, the total length of the exposed surface length per rotation was the same. Therefore, the volatile concentration was insensitive to the screw pitch.

The effect of the equilibrium concentration is shown in Figure 10c. The equilibrium concentration determines the capacity of the vacuum pump. If a high-capacity vacuum pump is used, the equilibrium concentration will be smaller. As expected by Equation (10), the volatile concentration increases with the equilibrium concentration. A high equilibrium concentration indicates a low capacity, i.e., a low flow rate of the vacuum pump. The low-capacity pump results in a high volatile concentration.

## 5. Conclusions

We developed a calculation method for determining the devolatilization efficiency and volatile-component concentration using Latinen’s surface renewal model, where the exposed surface boundary length is given by the local degree of fill in the self-wiping corotating twin-screw extruder. The local degree of fill was obtained from the method described in our previous study, which was used to determine the resin profile of starved screws. The concentrations of toluene as a volatile component in the simulation and experiments at the exit of the corotating twin-screw extruder agree well. The increase in the screw rotation speed increases both the exposed surface boundary length and the devolatilization efficiency. The flow rate and pump capacity are effective in increasing the devolatilization efficiency and decreasing the volatile concentration. The increase in the screw pitch of a full-flight screw is insensitive to the devolatilization efficiency.

However, our simulation method has several drawbacks as well. Our calculation method of local degree of fill cannot be applied to extruders such as a counterrotating twin-screw extruder. The flow calculation to determine the local degree of fill is decoupled with the devolatilization efficiency calculation. The viscosity of the resin is a function of the volatile component concentration. The viscosity increases with increasing devolatilization. The viscosity change affects the pressure distribution, the degree of fill, and finally, the devolatilization efficiency. In addition, it is challenging to incorporate the viscosity change in the devolatilization simulation. This study used the surface renewal model of devolatilization, which is only one type of devolatilization. We will investigate the incorporation of foam devolatilization, where bubble nucleation and growth occur, in our next study.

## Figures and Tables

**Figure 1 polymers-12-02728-f001:**
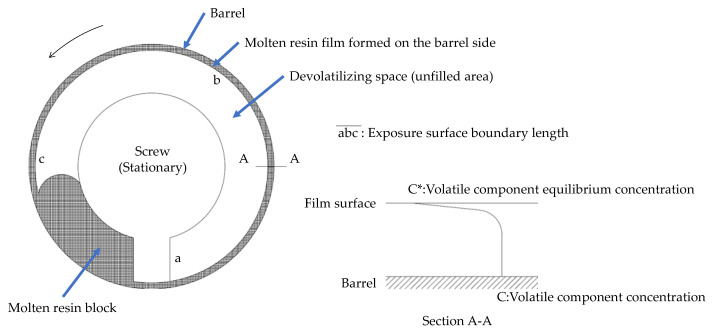
Schematic illustration of Latinen’s model for devolatilization in a single-screw extruder.

**Figure 2 polymers-12-02728-f002:**
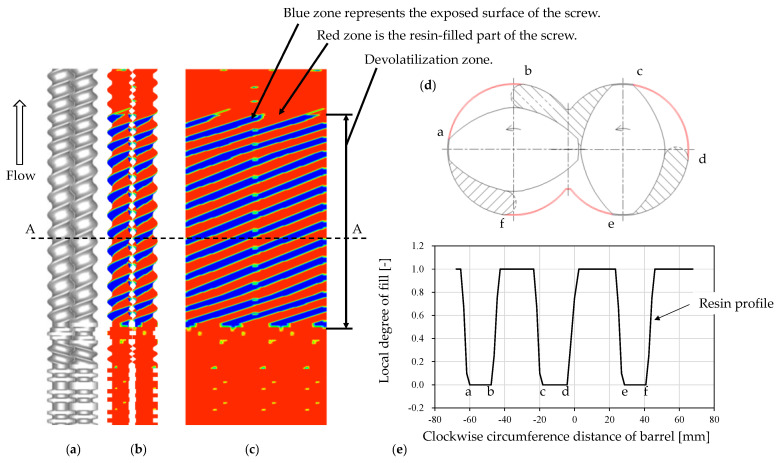
Devolatilization calculation section (*z*_e_–*z*_s_) and the exposed surface boundary length in a twin-screw extruder. (**a**) Screw configuration. (**b**) Resin distribution: red and blue indicate the local degree of fill 1 and zero, respectively. (**c**) Resin distribution on unwrapped screw geometry. (**d**) Cross-sectional view of screw, barrel, and resin in the starved zone. (**e**) Local degree of fill distribution as a function of clockwise circumference distance of the barrel.

**Figure 3 polymers-12-02728-f003:**
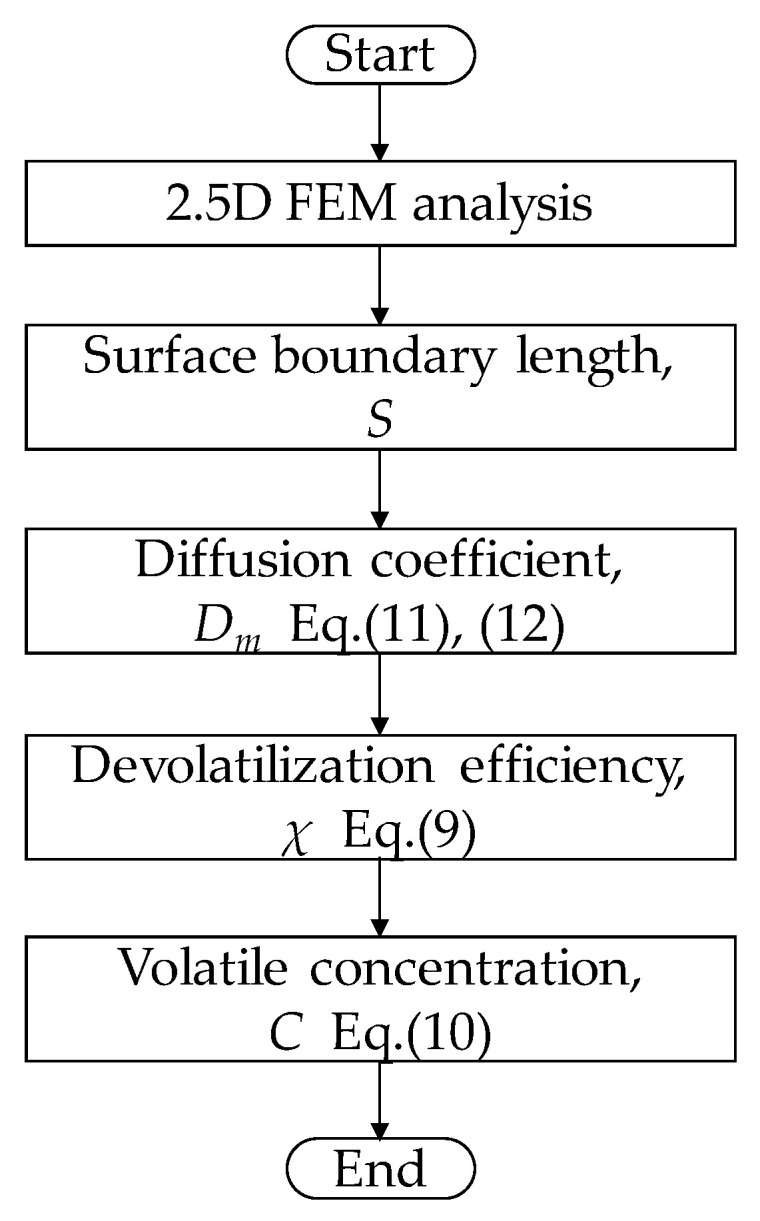
Flow chart for calculating the volatile concentration.

**Figure 4 polymers-12-02728-f004:**
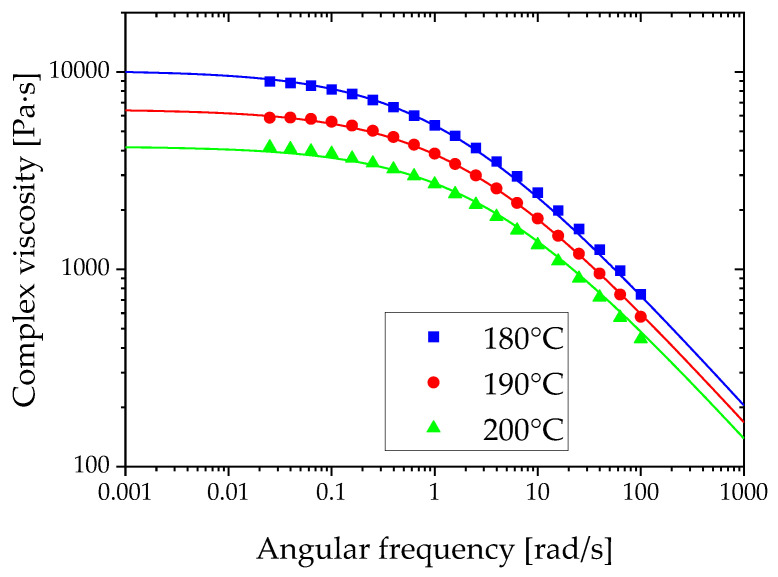
Viscosity data and curves obtained using the Cross model.

**Figure 5 polymers-12-02728-f005:**
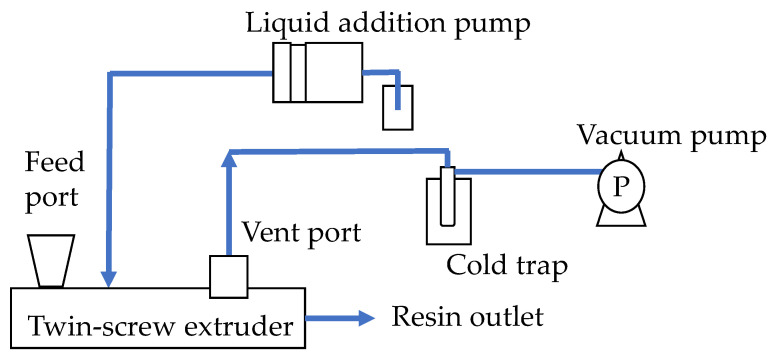
Schematic diagram of the apparatus configuration.

**Figure 6 polymers-12-02728-f006:**
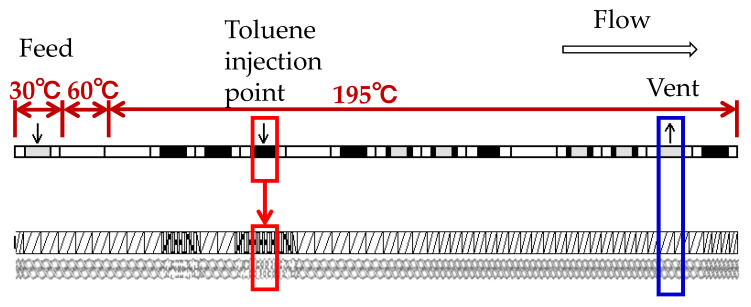
Screw configuration and barrel temperature settings.

**Figure 7 polymers-12-02728-f007:**
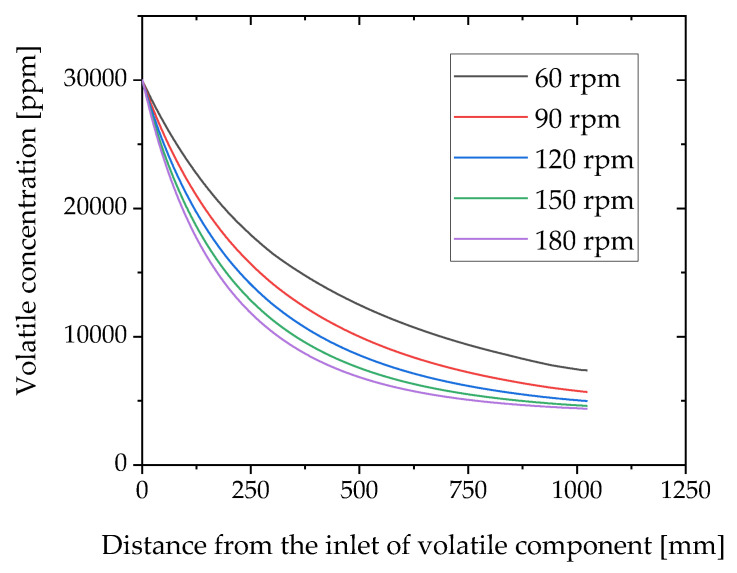
Simulation results of toluene concentration along the machine direction of twin-screw extruder.

**Figure 8 polymers-12-02728-f008:**
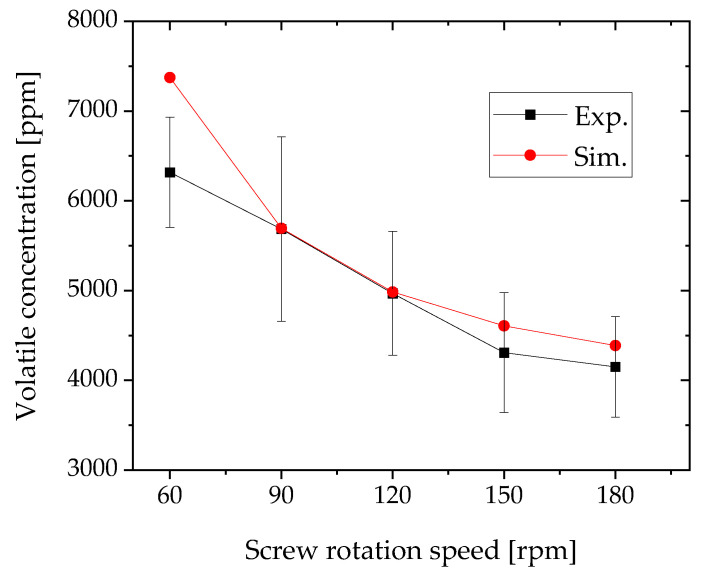
Comparison of simulation results and experimental results of screw rotation speed and toluene concentration (starved zone length = 1024 mm). The volatile concentration at each screw rotation speed was measured three times.

**Figure 9 polymers-12-02728-f009:**
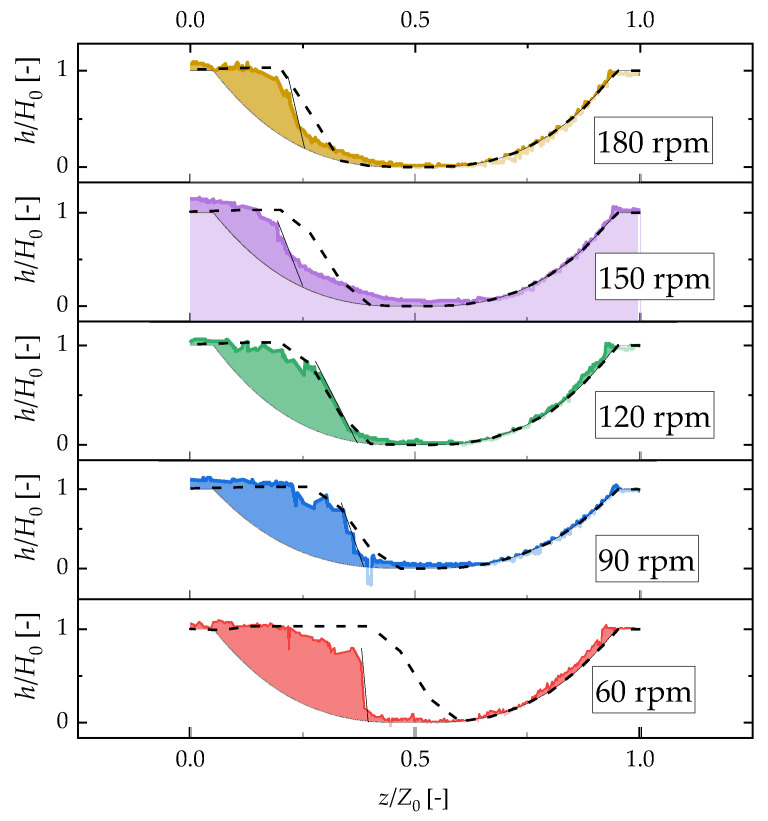
Effect of screw rotation speed on the resin distribution of the experimental and simulation results. The dotted lines show the simulation results and the experimental results are filled with colors.

**Figure 10 polymers-12-02728-f010:**
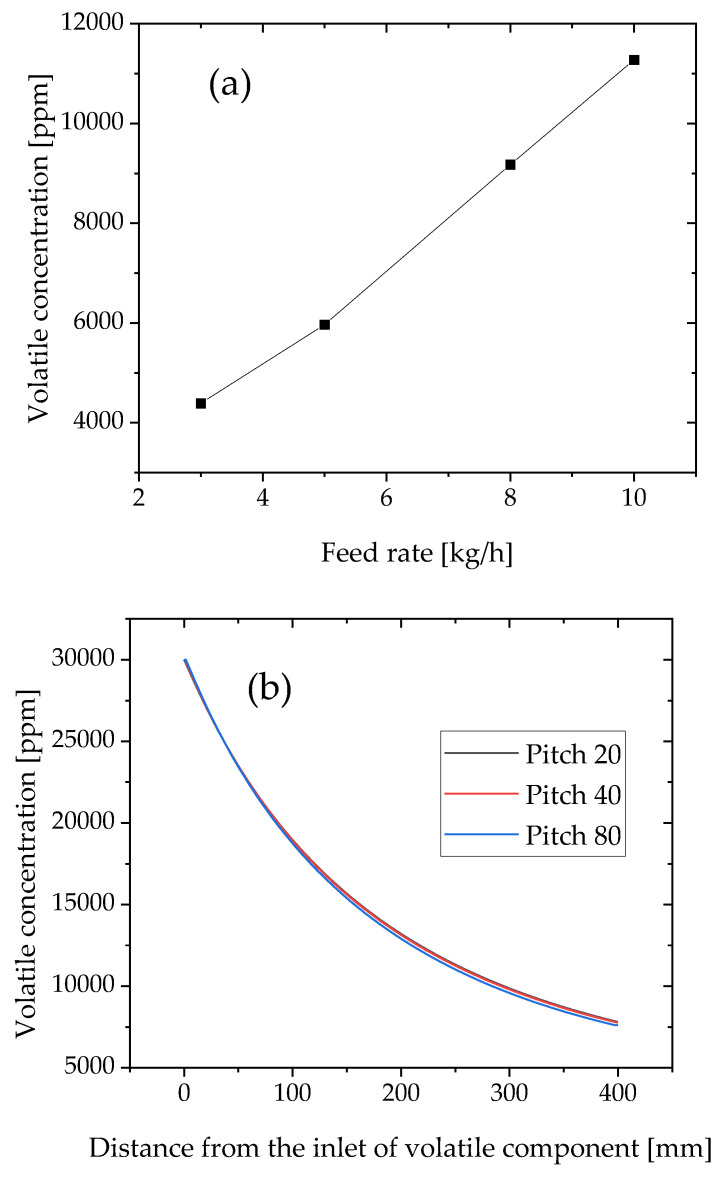
Simulation result for the effects of (**a**) flow rate, *Q*, (**b**) screw pitch, and (**c**) equilibrium concentration on the concentration of volatile components in the extrudate.

**Figure 11 polymers-12-02728-f011:**
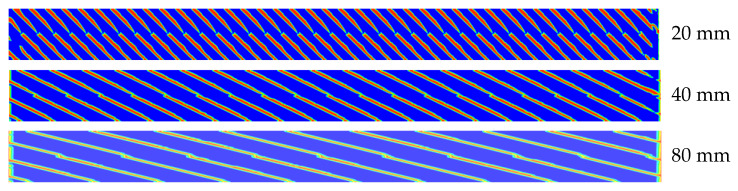
Degree of fill distribution of different screw pitches.

**Table 1 polymers-12-02728-t001:** Diffusion coefficient of polypropylene-toluene.

Temperature (°C)	Diffusion Coefficient (m^2^/s)
25	$4.58\times 10^{-13}$
50	$3.44\times 10^{-12}$
80	$1.16\times 10^{-11}$

**Table 2 polymers-12-02728-t002:** Parameters used to calculate the diffusion coefficient.

Initial Volatile Concentration of Experiment (ppm)	Equilibrium Volatile Concentration, *C** (ppm)	α	β	*D*_r_ (m^2^/s)	*T*_r_ (°C)
30,000	4000	0.04	1	$4.58\times 10^{-13}$	25

**Table 3 polymers-12-02728-t003:** Parameters of the Cross model.

Symbol	Value	Unit
*A*	8.48 × 10^−6^	Pa∙s
*T_b_*	9470	K
*C*	0.421	–
Τ*	12,300	Pa

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
