# Peer review of "Experimental and Numerical Simulation Study of Devolatilization in a Self-Wiping Corotating Parallel Twin-Screw Extruder"

_polymers, 2020, doi:10.3390/polym12112728_

Round 1

Reviewer 1 Report

Devolatilization is an important process in the production of polymers using twin-screw extruders. The authors used the numerical simulation method to calculate the surface area and residence time of devolatilization. The  numerical results agree well with the experimental ones. The comparison between the numerical and experimental studies is significant, and the results are interesting. I recommend publication of the manuscript after the following minor revisions: the numerical calculation method should be explained in detail in the manuscript, which may help the reader to understand the results.

Author Response

Dear Reviewer1,

Thank you for reviewing our manuscript. We have revised the manuscript in accordance with your comments. I ask you to review our manuscript again. We hope that the revised manuscript is now suitable for publication.

We have added flowchart and comments for the numerical calculation method.

However, the details are omitted because they are described in our paper in the reference {14.Ohara, M.; Tanifuji, S.-i.; Sasai, Y.; Sugiyama, T.; Umemoto, S.; Murata, J.-i.; Tsujimura, I.; Kihara, S.-i.; Taki, K. Resin distribution along axial and circumferential directions of self-wiping co-rotating parallel twin-screw extruder. AIChE J, 2020, e17018.}.

Sincerely regards,

Department of Natural Science, Graduate school of Kanazawa University, Japan.

Masatoshi Ohara

Reviewer 2 Report

Overall, it is a well-organized and written manuscript. There are only several minor comments that require the authors to address.

Abstract – Authors need to highlight the key findings (in value) of this work in the abstract

Introduction – Ref 11, 12 and 13 are missing from the main text. Ref 14 was quoted directly after Ref 10. Besides, authors need to include more recently published works to show the current development of this research topic.

Objective should be clearly stated at the end of Introduction

Figure 3 – Colour should be applied to make the figure more presentable.

Table 4 – I strongly advise this table to be arranged as Supplementary Data

Figure 6 – Authors need to explain why such range (60-180 rpm) was chosen and studied in this work.

Figure 7 – The standard deviation of each data is very large. What is the reproducibility of the data? Please explain in detail.

Conclusion – Key findings of this work should be clearly mentioned in Conclusion.

Author Response

Dear Reviewer2,

Thank you for reviewing our manuscript. We have revised the manuscript in accordance with your comments. I ask you to review our manuscript again. We hope that the revised manuscript is now suitable for publication.

Abstract – Authors need to highlight the key findings (in value) of this work in the abstract

We have modified the abstract to clarify the motivation of this study and results.

Introduction – Ref 11, 12 and 13 are missing from the main text. Ref 14 was quoted directly after Ref 10. Besides, authors need to include more recently published works to show the current development of this research topic.

We have added the missing reference in line 68, p.2.

The recent research have been also added in line 91, p.2.

Objective should be clearly stated at the end of Introduction

We have modified the end of introduction and added a paragraph for explaining each section.

Figure 3 – Colour should be applied to make the figure more presentable.

We have modified Figure 3.

Table 4 – I strongly advise this table to be arranged as Supplementary Data

We have moved Table 4 to Appendix.

Figure 6 – Authors need to explain why such range (60-180 rpm) was chosen and studied in this work.

We have commented the reason why the range of screw rotational speed was chosen in line 295-298, p.10. For higher rotational speeds, the resultant volatile concentration would not almost change.

Figure 7 – The standard deviation of each data is very large. What is the reproducibility of the data? Please explain in detail.

In Figure 7, the experimental data were measured 3 times under each rotational speed. We guess that the reason for large standard deviation of volatile concentration is because the fluctuation of the surface area in the twin-screw extruder was large. This fact was already reported in our previous paper, (see Figure7 in Ref. [17]).

However, there is a tendency for the toluene concentration to decrease as the screw rotation speed increases.

It would be improved by increasing the number of measurements.

Conclusion – Key findings of this work should be clearly mentioned in Conclusion.

We think the key findings are already mentioned in conclusion:

  1. We have developed the calculation method for volatile concentration in self-wiping corotating twin screw extruder using the Latinen’s model combined with 2.5D FEM analysis.
  2. The experiments for examining the simulation results have been done and the experimental results agreed well with the simulation results.
  3. The flow rate and pump efficiency were found to be effective for devolatilization, but screw pitch was insensitive.

Sincerely regards,

Department of Natural Science, Graduate school of Kanazawa University, Japan.

Masatoshi Ohara